# Human Papillomavirus 16 E6 Suppresses Transporter Associated with Antigen-Processing Complex in Human Tongue Keratinocyte Cells by Activating Lymphotoxin Pathway

**DOI:** 10.3390/cancers14081944

**Published:** 2022-04-12

**Authors:** Ati Burassakarn, Pensiri Phusingha, Takashi Yugawa, Kazuma Noguchi, Tipaya Ekalaksananan, Patravoot Vatanasapt, Tohru Kiyono, Chamsai Pientong

**Affiliations:** 1Department of Microbiology, Faculty of Medicine, Khon Kaen University, Khon Kaen 40002, Thailand; atibura@kku.ac.th (A.B.); tipeka@kku.ac.th (T.E.); 2HPV & EBV and Carcinogenesis Research Group, Department of Microbiology, Faculty of Medicine, Khon Kaen University, Khon Kaen 40002, Thailand; patvat@kku.ac.th; 3Center of Excellence for Antibody Research (CEAR), Faculty of Tropical Medicine, Mahidol University, Bangkok 10400, Thailand; pensiri.phu@mahidol.ac.th; 4Division of Carcinogenesis and Cancer Prevention, National Cancer Center Research Institute, Tokyo 104-0045, Japan; tyugawa@ncc.go.jp; 5Department of Oral and Maxillofacial Surgery, Hyogo Medical University, Mukogawa-Cho 1-1, Nishinomiya 663-8501, Japan; knoguchi@hyo-med.ac.jp; 6Department of Otorhinolaryngology, Faculty of Medicine, Khon Kaen University, Khon Kaen 40002, Thailand; 7Project for Prevention of HPV-Related Cancer, Exploratory Oncology Research and Clinical Trial Center, National Cancer Center, 6-5-1 Kashiwanoha, Kashiwa 277-8577, Japan

**Keywords:** HPV16 E6, oral squamous cell carcinoma, immune evasion, lymphotoxin, TAP

## Abstract

**Simple Summary:**

There is still limited knowledge of the critical pathogenic processes by which HPV16 induces oral carcinogenesis. Therefore, we aimed to illuminate the oncogenic role of HPV16 in the context of oral squamous cell carcinomas (OSCCs). Using human tongue keratinocyte cells, we demonstrated that HPV16 E6 promotes LTα_1_β_2_ and LTβR expression, thus promoting the lymphotoxin signaling pathway and leading to suppression of the transporter associated with the antigen-processing complex (TAPs; TAP1 and TAP2). Additionally, in vitro, we also demonstrated regulation of the antigenic peptide-loaded machinery in HPV-infected OSCC tissues through analysis of the transcriptomic profiles of the head and neck squamous cell carcinoma (HNSCC) cohort from the TCGA database, which was validated using fresh biopsied specimens. Thus, our study enhances the proposed functional role of HPV16 E6-associated immune-evasive properties in oral epithelial cells, revealing a possible mechanism underlying the development of HPV-mediated OSCCs.

**Abstract:**

Infection by high-risk human papillomaviruses (hrHPVs), including HPV type 16 (HPV16), is a major risk factor for oral squamous cell carcinomas (OSCCs). However, the pathogenic mechanism by which hrHPVs promote oral carcinogenesis remains to be elucidated. Here, we demonstrated that the suppression of a transporter associated with the antigen-processing complex (TAPs; TAP1 and TAP2), which is a key molecule in the transportation of viral antigenic peptides into MHC class-I cells, is affected by the E6 protein of HPV16. Mechanistically, HPV-mediated immune evasion is principally mediated via the signal-transduction network of a lymphotoxin (LT) pathway, in particular LTα_1_β_2_ and LTβR. Our analysis of transcriptomic data from an HNSCC cohort from the Cancer Genome Atlas (TCGA) indicated that expression of TAP genes, particularly TAP2, was downregulated in HPV-infected cases. We further demonstrated that LTα_1_β_2_ and LTβR were upregulated, which was negatively correlated with TAP1 and TAP2 expression in HPV-positive clinical OSCC samples. Taken together, our findings imply that HPV16 E6 regulates the machinery of the antigenic peptide-loading system and helps to clarify the role of oncogenic viruses in the context of oral carcinoma.

## 1. Introduction

Cancers in the oral cavity are recognized as one of the leading cancers, accounting for approximately 2% of all cancers worldwide and having a 50% mortality rate [1]. The most common type is squamous cell carcinoma (SCC), which arises in the tongue and the floor of the mouth [2]. The use of tobacco, alcohol consumption and areca-nut chewing are well-defined factors involved in the initiation of these carcinomas. However, over the past decades, several reports have revealed an increasing prevalence of human papillomaviruses (HPVs) in oral cancer cases [3,4].

Protein p53 is degraded by hrHPV E6 while E7 targets the retinoblastoma (Rb) protein and its family members [5,6], leading to inappropriate cell cycle progression in HPV-infected cells [7,8]. However, HPV by itself is not sufficient to induce epithelial cell transformation. The long-established theory of immune surveillance asserts that the immune system counteracts the growth of abnormal cells by recognizing and eliminating the vast majority of initial cancer cells and thus nascent tumors [9]. Multiple lines of evidence have suggested that HPV oncogenes contribute to evasion of the cell-mediated immune response (CMIR), thus leading to persistent HPV infection and increased risk of SCC development [10,11]. Most of the studies in this field focus on the alteration of the antigen-presentation pathway i.e., the reduction of MHC-I expression by hrHPV E6 [12,13] or E7 [14,15,16,17]. 

The effective clearance of viral infections and abnormal transformed cells by the CMIR is orchestrated by the antigen-processing machinery. The key molecules in this process are the transporters associated with antigen processing (TAPs). TAP1 serves the targeted peptides to the MHC-I molecules via the peptide-loading complex and carries the antigenic peptides from the cytoplasm into the endoplasmic reticulum [18,19]. Furthermore, TAP1 is essential for the formation and stability of MHC-I loaded with antigenic peptides [20,21]. The expression of HPV16 E7 interferes with the IFN-γ/JAK-STATs/IRF-1 signaling pathway leading to suppression of TAP1 expression, which in turn affects the efficiency of peptide loading and MHC class I antigen presentation in mouse keratinocyte cells [22]. The loss of expression of MHC class I and TAP1 allow some cancer cells to evade immune surveillance and contribute to the clinical course of OSCC and other cancers [23,24].

Lymphotoxin (LT) is a cytokine related to the tumor necrosis factor superfamily and exists in two molecular forms, LTα_3_ (secreted form) and LTα_1_β_2_, a membrane-associated heterotrimeric form, which binds to tumor necrosis factor (TNF) receptors and LTβ receptor (LTβR), respectively [25]. Interestingly, LT and TNF-α genes are tandemly localized within the HLA locus [26]. The network of LTα_3_–LTα_1_β_2_ provides a connection between lymphoid cells and the microenvironment of cells. The different ligand–receptor interactions can stimulate the pathways of pro-inflammatory (i.e., LTα_3_–TNFR1) or homeostatic signaling (i.e., LTα_1_β_2_–LTβR). In the adult, LTα_1_β_2_ in B cells provides key differentiation signals to LTβR-expressing stromal cells and macrophages for initiation of type 1 interferon production targeting viral pathogens [27,28]. Additionally, the formation of tertiary lymphoid tissue at extra-lymphatic sites of chronic inflammation is orchestrated by LTα_1_β_2_. Importantly, frequent mutations of signaling pathway components of the TNF receptor superfamily (ubiquitin E3 ligases, TRAF3, TRAF2, and cIAP), regulating NF-kB-inducing kinase (NIK), have been reported in several cancers such as cervical cancer, prostate cancer and liver cancer [29].

Another role of LT networks, i.e., LTβR signaling, is involved in the progression of cancer by inhibiting the adaptive immune response of the host to cancer cells. In 2011, Kim and colleagues [13] demonstrated that the expression of LTα, LTβ, and LTβR was upregulated by HPV16 E6 oncogene in cervical cancer cell lines. Interestingly, the signal transduction from LTβR downregulated MHC class I expression and promoted these cells’ resistance to cytotoxic T lymphocyte-mediated lytic activity. 

High-risk HPV types including HPV16, 18, and 58 are found in more than 90% of all HPV-positive OSCCs [30]. The involvement of HPV in OSCC progression has been investigated in various studies but the mechanism by which the virus promotes the development of OSCC remains inconclusive [31]. Therefore, we aimed to elucidate the association of HPV with oral carcinogenesis. Here, using HPV-positive human tongue keratinocyte cell lines, we identify a molecular pathway by which HPV16 E6 mediates suppression of TAP2, possibly through signaling of the LT network. This network plays an important role in OSCC development by affording evasion of the host immune-defense mechanism.

## 2. Materials and Methods

### 2.1. Ethical Statement

Informed consent was obtained from all participants, authorizing access to their clinical information and tumor samples for research purposes. All procedures performed in studies involving human participants were approved by the Khon Kaen University Ethics Committee in Human Research (HE521344 and HE581211) and were according to the 1964 Helsinki Declaration.

### 2.2. Clinical Specimen of OSCCs

Both fresh biopsies and archived formalin-fixed paraffin-embedded (FFPE) tissues were used. Twenty-three OSCC tissue biopsies were freshly excised from the marginal tumor bed by an otolaryngologist and were histologically confirmed as squamous cell carcinoma of the tongue, lip, the floor of the mouth, and other parts of the mouth (C00-C06), according to the codes from ICD-O-3. All patients had been residents in northeastern Thailand for more than five years before enrollment. In addition to the fresh biopsies, FFPE tissues from 146 patients with different tumor grades of OSCC, diagnosed at three hospitals in northeastern Thailand (Srinagarind Hospital, Khon Kaen Hospital, and Sunpasitthiprasong Hospital) were investigated. These collections were approved by the Khon Kaen University Ethics Committee in Human Research (HE 521344). Appendix A shows the clinico-demographic information relating to these FFPE specimens. 

### 2.3. Cell Lines and Cultures

The human tongue keratinocyte cell line HTK1 [32], and HTK1 transduced with either LXSN-16E6E7, LXSN-18E6E7, LXSN-16E6SD, or empty retrovirus vector, LXSN, were used in this study. These cell lines were cultured in the optimized in-house keratinocyte medium consisting of F-medium [32] supplemented with 10 μM ROCK inhibitor, Y-27632 (Select, Houston, TX, USA), 500 nM TGF-β receptor inhibitor, A83-01 (Tocris Bioscience, Bristol, UK), and 5% (*v*/*v*) conditioned medium containing Wnt3a and R-Spondin-1 [33]. HNSCC-derived HPV-negative cell lines including, HSC1, HSC2, HSC3, SCCKN, SCC25, SCCTH, UM-SCC-14A, and the HPV16-positive cell line derived from SCC of the tongue, UPCI: SCC90 (abbreviated to SCC90 in results), along with two HPV-positive cervical cancer cell lines, SiHa (HPV16) and HeLa (HPV18), were used in the functional analyses. Both HNSCC and cervical cancer cell lines were maintained in 10% fetal bovine serum (FBS; Gibco, ThermoFisher, Waltham, MA, USA) supplemented with Dulbecco′s Modified Eagle′s medium (DMEM; Nacalai Tesque, Kyoto, Japan). Reverse-transcriptase polymerase chain reaction (RT-PCR) was implemented to verify the HPV status of all cell lines used in downstream experiments by detecting the presence of E6/E7 mRNA. Appendix A shows the origin of cell lines used in this study.

### 2.4. Measurement of Gene Expression by Quantitative RT-PCR (qRT-PCR)

Total RNA from the cell lines was extracted using the RNeasy Mini Kit (Qiagen, Hilden, Germany) according to the manufacturer’s instructions. Briefly, the mixture of RNA lysate and 70% ethanol was applied to the RNeasy spin column and centrifuged at 10,000× *g* for 15 s. After washing the spin column with Buffer RW1 and Buffer RPE, 50 μL RNase-free water was added to the spin column membrane, and then RNA was eluted by centrifugation at 10,000× *g* for 1 min. For the fresh tissue biopsies, TRIzol^TM^ reagent (Invitrogen, Milan, Italy) was used for total RNA extraction according to the manufacturer’s instructions. The tissue samples were homogenously mixed with TRIzol^TM^ reagent. After the addition of chloroform, the mixtures were centrifuged at 12,000× *g* for 15 min at 4 °C. RNA remained exclusively in the aqueous phase. RNA was pelleted and washed using isopropyl alcohol and 75% ethanol, respectively. Total extracted RNA was stored at −80 °C until use. 

The extracted RNA was reverse-transcribed to cDNA using PrimeScript^TM^ RT reagent Kit (TaKaRa, Shiga, Japan). According to the manufacturer’s protocol, a total of 20 µL of reagent mix was prepared, including 4 µL of 5x PrimeScript buffer, 1 µL PrimeScript RT Enzyme Mix, 1 µL Oligo dT Primer (50 µM), 1 µL random 6 mers (100 µM), 1 µL of 1 µg RNA samples and 12 µL RNase Free dH_2_O. The mixture was incubated at 37 °C for 15 min followed by 85 °C for 5 s to heat-inactivate the reverse transcriptase, cooled down at 4 °C and then cDNA products were stored at −20 °C until use. Quantification of mRNA was done using a PrimeScript^TM^ RT reagent Kit (TaKaRa, Shiga, Japan) in a 10 µL volume including 5 µL KAPA SYBR qPCR Master Mix, 0.4 µL of each primer, 100 ng of RT product, and 3.2 µL PCR-grade water. The mixture was pre-denatured at 95 °C for 20 s and 40 cycles of 95 °C for 30 s followed by 60 °C for 30 s. The melting curve was at 60 °C for 1 min to 95 °C for 15 s. All samples were analyzed in duplicate using an Applied Biosystems StepOnePlus Real-time PCR System (Applied Biosystems, Waltham, MA, USA). Appendix A shows oligonucleotide sequences that were used in this study.

### 2.5. Determination of Protein Concentrations Using Western Blotting

This assay was executed as previously described by Yugawa and colleagues [34]. Briefly, whole cells were lysed in WE16th lysis buffer [35] and then sonicated. The lysates were centrifuged. Total protein concentration was measured using the DC^TM^ protein assay (Bio-Rad, Hercules, CA, USA). Samples, each containing 20 μg of protein, were electrophoresed under reducing conditions in SDS-polyacrylamide gels of appropriate percentage. Protein was electrically transferred to PVDF membranes (Amersham Biosciences, Buckinghamshire, UK) at 100 mA for 70 min. After blocking with 5% skim milk solution, the blots were probed with an optimal concentration of primary antibodies. The membranes were washed with 1xTBS-T buffer and incubated with the horseradish peroxidase-conjugated secondary antibody. Bound antibodies were detected with the enhanced-chemiluminescence Western blotting substrate (Roche, Basel, Switzerland). Our blots were also probed for total vinculin as a loading control. Appendix A shows the list of antibodies used in this study. The original western blotting results were shown in Appendix A.

### 2.6. Gene Silencing by Retroviruses Expressing Short Hairpin RNAs (shRNA)

Retrovirus (pSI-MSCVpuro-H1R) particles carrying either shLTB or shLTBR and mock-infected controls were constructed using the Gateway recombination cloning system (Invitrogen, Waltham, MA, USA) as previously described [36]. HEK-293T cells were used for the production of these constructed retroviruses. Retroviral transductions were performed on HTK1-16E6SD, HTK1-LXSN, and SCC90 cell lines following a standard protocol. Briefly, the listed cell lines were cultured on 6-well plates for 24 h before infection. The retroviruses carrying shRNA were added into each culture well at the multiplicity of infection of 3 in the presence of 4 µg/mL of polybrene. After 24–30 h, cells were selected in the presence of 1 µg/mL of puromycin until the mock-infected cells completely died. The expression of LTB and LTBR was subsequently detected by quantitative RT-PCR. Either LTβ- or LTβR-silenced cells were cultured and used in the downstream experiments (Appendix A).

### 2.7. siRNA and Cell Transfection

The sequences of HPV16E6 were obtained from GenBank (NC_001526.4). The public software Dharmacon™ (Lafayette, CO, USA; https://horizondiscovery.com, 17 September 2021) was used to design sequences of E6-specific siRNA. The outputs were aligned with E6 mRNA gene sequences and the highest similarity score was selected (Appendix A). For the transfection, the optimal cell density was seeded in 24-well plates. The siRNA sequences targeting HPV16 E6 were transfected using Lipofectamine^®^ RNAiMAX Reagent (Invitrogen, Waltham, MA, USA) at a final concentration of 50 nM, according to the manufacturer’s instructions. The efficiency of transfection was determined after 6 h relative to FITC-labeled siRNA negative control using a fluorescence microscope (ZEISS Axioscope 5, Oberkochen, Germany). 

### 2.8. Analysis of RNA Sequence Data from TCGA

We retrieved the level 3 RNA-Seq data of the TCGA head and neck cancer cohorts from the Broad Genome Data Analysis Center’s Firehose server (https://gdac.broadinstitute.org/, accessed on 3 June 2021). For the analysis of gene expression, the Expectation-Maximization (RSEM) normalized Illumina HiSeq RNA expression data of the targeted gene were obtained from primary patient samples with known HPV status that were classified as HPV+, HPV−, or normal control tissue. We omitted the calculation of indicated gene expression from patient samples with unknown HPV status and from secondary metastatic lesions [37]. This resulted in data being available from 48 HPV+, 293 HPV−, and 44 normal control samples. 

### 2.9. Tissue Microarray (TMA) Construction

All OSCC FFPE tissue samples were sectioned at 5 μm thickness using a microtome and stained with hematoxylin and eosin (H&E) and examined independently by two pathologists to identify an actual tumor area. From the positive tumor area in the FFPE block, a core was punched using a 0.6 mm diameter stainless-steel tube and transferred into a TMA mold. TMA tumor tissue blocks were sectioned at 5 μm thickness using a microtome to prepare for use in the downstream experiments.

### 2.10. Detection of HPV DNA by Polymerase Chain Reaction (PCR)

DNA was extracted from oral tissue specimens using a DNeasy^®^ Blood & Tissue Kit (QIAGEN, Hilden, Germany) according to the manufacturer’s instructions. Briefly, five tissue sections in a 1.5 microcentrifuge tube were vigorously mixed with xylene and centrifuged at full speed at room temperature (RT) for 2 min. After removing the supernatant, the cell pellet was washed with absolute ethanol. The pellet was resuspended in ATL buffer containing proteinase K at 56 °C until complete lysis. Total RNA was degraded by adding RNase A solution. AL buffer and ethanol were added and mixed. The entire lysate was carefully transferred to a QIAamp MinElute column and centrifuged at 6000× *g* for 1 min. After washing with AW1 and AW2 buffers, the column was placed in a clean 1.5 mL microcentrifuge tube and incubated with ATE buffer at RT for 5 min. Finally, the DNA was eluted by centrifugation at 20,000× *g* for 1 min. The integrity of the extracted DNA samples was confirmed by amplifying the *β-actin* gene (Appendix A).

The extracted DNA was reused for HPV DNA detection by HPV L1-specific PCR, which is accomplished using primers GP5+/GP6+ (Appendix A). The PCR reaction mixture contained 25 μL of 1x PCR buffer, 0.2 mM of dNTPs, 1.5 mM MgCl_2_, 0.1 mM of each primer and 0.25 units Taq DNA polymerase. The PCR conditions were initial denaturing at 95 °C for 5 min followed by 40 cycles, each of 95 °C for 1 min, 42 °C for 1 min, and 72 °C for 30 s, with a final extension at 72 °C for 4 min. The reaction mixture was separated through 2% agarose gel and stained with ethidium bromide, and subsequently visualized under an ultraviolet transilluminator. 

### 2.11. Detection of High-Risk HPV in OSCC Tissues by In-Situ Hybridization (ISH)

As described in our previous publication [38], the RNA ISH to detect high-risk HPV E6/E7 mRNA was manually performed using the RNAscope^®^ HPV kit (Advanced Cell Diagnostics Inc., Newark, CA, USA) according to the manufacturer’s instructions. To ensure uniformity of results, all TMA tissue samples were processed in parallel with positive, *ubiquitin C* (*UBC*), and negative controls, *dapB*. The specific positive-staining signals were seen as brown diffuse or punctate dots present in the cytoplasm and/or nucleus of the infected cells. The HPV status of each sample was scored qualitatively as either positive or negative, using *dapB*-stained slides as reference.

### 2.12. Immunohistochemistry (IHC) Assay

Paraffin wax was removed from the FFPE sections using xylene. After deparaffinization, FFPE sections were rehydrated in an ethanol concentration gradient. Antigen was retrieved by boiling in 10 mM sodium citrate buffer, pH 6.0 for 20 min. Endogenous peroxidase activity was blocked by 3% hydrogen peroxide (H_2_O_2_). After washing, the primary antibody (Appendix A) was applied to the tissue slide at RT for 90 min. A secondary antibody conjugated with HRP was added to the sections and incubated for 30 min. Detection was achieved using DAB solution (Tris-base, pH 7.4 with 3% H_2_O_2_ and 1% of DAB chromogen). Then, the samples were incubated with hematoxylin solution for 30 s to stain nuclei. The slides were mounted with a mounting medium. The signal of the targeted proteins was observed under a bright-field microscope. To assess the IHC score, a semiquantitative evaluation was performed. As described by Xiang et al., 2018 [39], scores of 0, 1, 2, 3, and 4 were considered when the staining in ≤1%, 2–25%, 26–50%, 51–75, and ≥75% of cancer cells, respectively. Moreover, the intensity of staining was also scored as the following criteria: 0 for no staining, 1 indicated weak staining, 2 represented moderate staining, and 3 designated strong staining. Staining was performed twice in the serialized sections from the same tissue samples that were independently examined by two pathologists. The staining index including numbers 0–12 was achieved by multiplying the intensity by the proportion of immunopositive cells of interest.

### 2.13. Systematic Review and Meta-Analysis

Our study was performed in line with Preferred Reporting Items for Systematic Reviews and Meta-Analysis (PRISMA) guidelines [40]. The 6 public databases including PubMed, EMBASE, Web of Science, LILACS, Scopus, and the Cochrane Library were searched from an unspecified start date until 9 January 2020 with language restrictions (English only) by using the following combined keywords and medical subject headings: (human papillomavirus OR HPV) AND (oral squamous cell carcinoma OR OSCC OR oral cancer). Depending on the title and abstract, all works were checked, and qualified reports were recollected for full-text review. Furthermore, the reference lists in each original and reviewed article were manually explored to avoid missing potential studies. As previously described by Rapado-González et al., 2020 [41] The original reports met the following criteria: (1) case-control studies of patients with oral cancer and healthy controls, (2) HPV DNA prevalence determined in tissue samples (FFPE or biopsies), and (3) sufficient data to calculate odds ratios (ORs) with 95% confidence intervals (CIs) were included. We excluded the works that were (1) in vitro or animal studies, (2) reviews, letters, personal opinions, book chapters, case reports, and conference abstracts, and (3) duplicate articles or suspicion of data overlap [41]. Two researchers (AB and PP) independently assessed each appropriate report and organized the data into a Microsoft Excel spreadsheet (Microsoft Corp., Redmond, WA, USA) with the following information: author, publication year, country, sample type, anatomical site of tumor, HPV detection method, number of samples, total HPV DNA and type-specific HPV DNA prevalence. 

### 2.14. Statistical Analysis

All experiments were carried out in triplicate, and at least three independent experiments were performed. The association between HPV status and OSCC was examined using Chi-square tests in SPSS software version 14.0 (SPSS Inc., Chicago, IL, USA). Data of mRNA expression were normalized relative to GAPDH gene expression. The results were analyzed and graphed using GraphPad Prism Software Inc. version 8.0 (Graphpad Software, Inc., San Diego, CA, USA). Statistical analysis was performed using t-tests, Mann–Whitney rank-sum tests, or one-way ANOVAs. A result was regarded as statistically significant if the *p*-value was <0.05. To assess the heterogeneity of our metanalysis, Cochran’s Q-based Chi-squared test and I^2^ statistics were performed. When the calculated I^2^ was more than 50% and/or the presence of a *p* < 0.10 for the Cochran’s Q test, the analysis was considered as having significant heterogeneity [41]. Based on the heterogeneity, the fixed or random-effects model of HPV prevalence DNA and HPV genotypes in oral cancer was calculated. The association between HPV16 DNA infection and OSCC risk was evaluated by pooled odds ratio (OR) and 95% confidence intervals (CIs) comparing the cases to other risks.

## 3. Results

### 3.1. HPV Infection Downregulated TAP1 and TAP2 Expression in OSCC Tissues

To elucidate the effect of HPV infection on the cell-mediated immune response in the oral carcinoma tissues, we first analyzed the RNA sequence data of OSCC cases in the HNSCC cohort from TCGA. Our analysis demonstrated that the expression of three classical MHC class-I genes (*HLA-A*, *HLA*-B, and *HLA-C*) was significantly higher in OSCC samples than in normal control tissues (Appendix A). We also determined the mRNA level of the genes involved in the MHC class-I peptide-loading complex consisting of *B2M*, *TAP1*, *TAP2*, *TABP*, *CANX*, *CALR*, *PDIA3*, *ERAP1*, and *ERAP2*. As indicated in Appendix A, all genes were expressed at higher levels in OSCCs when compared to the normal cases, except for *CANX*, which was expressed at a comparable level. Thus, HPV might regulate the host’s cell-mediated immune response to establish persistent infection in oral keratinocyte cells. To clarify this issue, we examined whether the expression of these genes involved in the cell-mediated immune response was lower in the HPV-positive OSCC subtypes compared to HPV-negative cases. In the HPV-positive OSCCs, the expression of *HLA-A*, *HLA-B*, *HLA-C*, *B2M*, *TAP1*, *TAP2*, *CALR*, and *PDIA3* was downregulated relative to the HPV-negative group (Figure 1A). On the other hand, HPV-positive cases exhibited a significant elevation of *ERAP1* and *ERAP2* genes compared with HPV-negative tissues (Figure 1A). Moreover, we found that there was no significant difference in the mRNA level of the *TAPBP* and *CANX* genes (Figure 1A). Since TAP plays a key role in the transportation of viral antigenic peptides into the endoplasmic reticulum (ER) and peptide loading on MHC class-I molecules, we next determined whether the expression of *TAP1* and *TAP2* was affected by HPV infection in OSCC cells. Using the qRT-PCR assay, our OSCC biopsies demonstrated the mRNA level of *TAP1* and *TAP2* was significantly lower in HPV-positive cases than in non-HPV infection cases (Figure 1B). Additionally, we also measured the *TAP1* and *TAP2* expression in both established OSCC cell lines naturally infected with HPV and uninfected cells. The qRT-PCR result showed that the expression of *TAP1* and *TAP2* genes was frequently suppressed in the HPV-infected OSCC cells (Figure 1C). At the mRNA level, the expression of TAP1 and TAP2 protein was downregulated by HPV infection (Figure 1D). Collectively, our analysis suggested that HPV might play an important role in the regulation of the antigenic peptide-loading machinery in OSCCs.

### 3.2. HPV16 Infection Is Associated with Increased Risk of OSCC

As the role of HPV in the suppression of TAP1 and TAP2 is implied by our results, we next sought to discover the specific HPV type that causes the suppression of such genes. Our previous study [38] reported that HPV16 (70.41%) was the most common genotype and was detected in both OSCC cases and control groups. Here, using the RNA in situ hybridization (RISH) assay we found E6/E7 HPV mRNA of HPV16 in 78.26% (72/92) of 168 FFPE specimens (Figure 2A). The expression of this mRNA can be observed as a dot-like nuclear pattern, indicating the transformative phase of HPV in OSCC cells (Figure 2B). 

We compared the distribution of hrHPV among OSCC patients from different Asian countries. The 11 studies represented a total sample size of 501 in the control group and 914 in the case group. The prevalence of HPV16 was higher among OSCC samples than in controls across all studies except for a study in Japan by Sugiyama et al., 2003 [42] (Figure 2C). An odds ratio (OR) and 95% CI were estimated to determine the association between HPV16 and OSCCs. Despite the results, high heterogeneity among the studies was marked due to the usage of various methodologies for identifying HPV in case and control groups. Of a combined 1813 tissue samples, 1052 patients were positive for HPV16, detected by various methods, and 761 cases were negative for HPV16. The calculated OR was 2.63 and 95% CI, 1.54–4.47, which showed a significantly increased risk of HPV among case groups when compared with the other risk factors (Figure 2D). Collectively, the results indicated that hrHPVs, particularly HPV16, are the most common HPV type and are associated with an increased risk of OSCCs.

### 3.3. HPV16 Oncogene-Mediated Suppression of TAP1 and TAP2 in Human Tongue Keratinocytes

E6 and E7 oncogenes of hrHPVs, including HPV16 and/or HPV18, are typically overexpressed in oral carcinomas [43]. Hence, we assessed the effect of both these viral oncogenes on *TAP1* and *TAP2* expression. Using the human tongue keratinocytes (HTKs), we exogenously expressed E6 and/or E7 from both HPV16 and HPV18 (Appendix A) and found that only HPV16 E6 caused the reduction of *TAP1* and *TAP2* expression in HTK cells (Figure 3A). Consistent with mRNA, immunoblotting indicated downregulation of both TAP1 and TAP2 protein in HTK-expressing HPV16 E6 (Figure 3B). These results indicate that E6 of HPV16 might play an important role as the regulator of TAP1 and TAP2 expression in HTK cells. To verify our results, we assessed the expression of these genes in both HPV16 E6-transduced HTKs and SCC90 with or without siHPV16 E6 treatment (Appendix A) using qRT-PCR and immunoblotting. As shown in Figure 3C–E, the depletion of HPV16 E6 restored the expression of TAP1 and TAP2 mRNA (Figure 3C,D) as well as their protein levels (Figure 3E) in HPV16 E6-expressing cells. Our findings highlight the ability of HPV16 E6 to downregulate TAP1 and TAP2 expression in human tongue epithelial cells.

### 3.4. LT Network Linked to TAP1 and TAP2 Suppression in HPV16 E6-Expressing Oral Epithelial Cells

Previous reports have indicated that HPV16 E6 can regulate the expression of LTα_1_β_2_ and its receptor, LTβR, inhibiting MHC class I expression [13]. Therefore, we speculated that the LT network might contribute to the HPV16 E6-mediated TAP1 and TAP2 suppression. To test this, the expression of LT-related genes was determined in OSCC cases of the HNSCC cohort from the TCGA database. As we expected, an increasing trend of LTα, LTβ, and LTβR could be found in the HPV-positive OSCCs when compared to the HPV-negative group (Figure 4A). Our qRT-PCR also verified the elevation of these genes in the HPV-positive OSCC biopsies compared to the cases without HPV infection (Figure 4B). Moreover, quantification of gene expression in OSCC biopsies by this assay also demonstrated the negative correlation between the *TAP1* and *TAP2* gene expression and the level of *HPV16 E6*, *LTα_1_β_2_*, and *LTβR* mRNA (Figure 4C). In addition to the mRNA expression assay, the upregulation of LT molecules was further confirmed by IHC staining. Using FFPE tissues, we found that some LT-related proteins, particularly LTβ, were upregulated in HPV-positive OSCC samples (Figure 4D). Together, these results suggest that HPV16 E6 might activate the LT signaling network, in particular, the LTα_1_β_2_–LTβR axis, contributing to TAP1 and TAP2 suppression in oral epithelial cells.

### 3.5. HPV16 E6 Regulates TAP1 and TAP2 Expression via LTα_1_β_2_-LTβR Axis in Oral Epithelial Cells

It has been reported that the activation of LTβR by its ligand, LTα_1_β_2_, affects the development of several cancers by inhibiting the adaptive immune response of host cells [44]. We next investigated whether the expression of TAP1 and TAP2 was affected by the LT network in HPV16 E6-expressing oral epithelial cells. In this experiment, we determined the expression levels of both LTβ and LTβR mRNA and protein in HPV16 E6-expressing HTK cells (HTK1-16E6). As shown in Figure 5A,B, the HTK1-16E6 cells expressed a high level of both LTβ and LTβR at the mRNA (Figure 5A) and protein levels (Figure 5B) compared to mock controls. Next, we designed and delivered the LTβ- and LTβR-specific shRNAs to HTK1-16E6 cells. This treatment potently suppressed the LTβ at both mRNA and protein levels (Figure 5C,D). Similarly, shLTβR-1, shLTβR-2, and shLTβR-3 inhibited the expression of LTβR at both mRNA and protein levels (Figure 5C,D). Using qRT-PCR and immunoblotting, the silencing of both LTβ and/or LTβR could restore the mRNA expression of TAP1 as well as TAP2 (Figure 5E,F). The protein level of TAP in the LT network-silenced HTK1-16E6 cells was higher than in control cells. (Figure 5G). These data suggested that the silencing of the LT network was able to restore antigenic peptide loading in oral epithelial cells.

## 4. Discussion

The heavy chain/light chain dimer loading with an antigenic peptide in the endoplasmic reticulum is important and required for the stable surface expression of MHC-I. The loading complex includes several crucial factors such as the peptide transporter system, TAP1/2, the bridging factor tapasin, and the chaperones (calreticulin, calnexin, and ERp57). Mutation or inhibition of these components can reduce the presentation of MHC-I-associated antigens, and lets pathogens or abnormal cells escape the recognition and clearance of T-lymphocyte [18,45]. 

The strongest evidence for human papillomavirus (HPV) involvement in epithelial tumorigenesis is in cervical cancer. However, increasingly, high-risk HPV (hrHPV), and especially infection with HPV16, is recognized as associated with OSCCs [36]. Several factors, including the host’s genetic background [46,47], environmental co-factors [48], and the ability of the virus to escape immune clearance [12,49] are required for the persistence of HPV and progression from acute HPV infection to cancer. As previously described, HPV can evade the host’s defense mechanisms indirectly by preventing access by immune cells to their proteins [50] and by direct blocking of the host antiviral response, particularly of the MHC-I antigen presentation to CD8^+^ effector T lymphocytes [51]. Although the downregulation of MHC-I expression is the major characteristic of HPV-associated cancer [14,52], the molecular mechanisms that bring this about remain to be determined. In this study, we found that the E6 oncoprotein of HPV16, but not of HPV18, dramatically decreased TAP1 and 2 protein levels. We further confirmed the correlation between HPV type and TAP expression in SCC90 cells which are HPV16-positive OSCC cells. We found that HPV16 can modulate both TAP1 and TAP2 expression. Taken together, the results support our hypothesis that the suppression of the immune response by a viral oncoprotein is through the downregulation of the antigen processing system.

Infection by HPV and the expression of HPV oncoproteins leads to alteration of host gene expression. Numerous previous studies have indicated that the E7 oncoprotein of high-risk HPV is involved in the evasion of the cell-mediated immune response. As described, this process occurs by transcriptional reduction of the MHC-I heavy-chain genes or blocking antigen loading and presentation [15,16,17]. Moreover, this mechanism has been proposed to explain the persistence of HPV infection and is necessary to allow HPV-associated cancers to escape the attention of CD8^+^ effector T-lymphocytes. Additionally, the third oncoprotein of high-risk HPV, E5, can also downregulate the expression of MHC-I. However, this is done at the non-transcriptional level in which the classical heavy chains are trapped in the Golgi apparatus and not transported to the cell surface [12].

Our analysis of 341 primaries human OSCC datasets, acquired from the HNSCC cohort in the TCGA database, indicated that HPV infection is a major factor associated with downregulation of the classical heavy chains of MHC-I and the peptide transporter system, TAP2 gene. Indeed, HPV-infected HNSCC exhibits significant increases, rather than decreases, in mRNA levels for many MHC-I components compared with normal control tissue. These unexpected results challenge the existing hypothesis of virally mediated immune evasion. However, an alternative pathway of activating MHC-I and components of the antigen-loading complex via IFN-γ-stimulating JAK/STAT/IRF-1 signaling cascades has been reported [22]. Furthermore, a higher level of IFN-γ mRNA has been found in HPV-infected samples than in non-HPV infection and normal control tissues in the TCGA database (data not shown). Moreover, the source of IFN-γ mRNA was determined to be from infiltrating lymphocytes (data not shown). Significant infiltration by T lymphocytes as well as by NK cells occurs in both HPV-positive and HPV-negative HNSCCs. Therefore, the high levels of mRNAs for MHC-I components in HPV-associated HNSCC provide evidence that oncoproteins of high-risk HPV are unable to efficiently block the production of IFN-γ at the transcriptional level in these carcinomas.

For the TCGA database, they conducted their analysis with the large numbers of tissue samples of head and neck cancer patients from various regional cancer centers, globally. Therefore, the differences in geography and ethnicity of patients are one explanation for the varied HPV-mediated LTα, LTβ, and LTβR upregulation. However, inconsistencies in the HPV prevalence in the oral cavity, in particular, HPV-positive oropharyngeal carcinomas, have obscured the ability to definitively associate HPV with OSCC. In addition, non-keratinized OSCC exhibiting high E6 and/or E7 mRNA expression shows a more increased LTα, LTβ, and LTβR expression than the others, suggesting that LTα, LTβ, and LTβR mRNA expression may be varied in assigning HPV-positive OSCC [13,53]. In Figure 4B, the expression of such genes was evaluated from our previous tissue OSCC samples that were retrieved from Southeast Asia’s patients and showed high E6/E7 mRNA levels [38]. Accordingly, it might be observed that HPV16 infection is statistically associated with an increased expression of MHC-I-dependent genes in OSCCs. However, additional high-quality studies with larger sample sizes are needed to further confirm such relationships.

Although LT was indicated to be a potent inducer of MHC-I expression in human endothelial cells [54,55] and human T-cell hybridomas [56], LT was largely powerless to control the expression of MHC [56]. Interestingly, the double-deficiency of TNF and LTα in mice leads to increased MHC-I expression in endothelial cells [57]. In association with NF-kB, the LT signaling might play an important role in the MHC-I downregulation. Our experiments revealed that an ectopic expression of HPV16 E6 in HTK cells strongly suppresses the expression of TAP1 and TAP2 concomitant with induction of the lymphotoxin network including LTα_1_β_2_ and LTβR, and that the silencing of this network by shRNAs restored the expression of TAPs, indicating that HPV16 E6 inhibits TAP gene expression via the signaling of the LTα_1_β_2_-LTβR network. Our data are in agreement with a previous report that the suppression of MHC-I gene expression in thyroid cells involves enhancer A and the transcription factor NF-kB [58].

## 5. Conclusions

This study proposed a novel tumorigenic function of HPV16 E6 in oral squamous-cell carcinomas which involved the suppression of cellular-orchestrated immunity in the tumor microenvironment of the oral cavity, allowing immune invasion and cancer development (Figure 6).

## Figures and Tables

**Figure 1 cancers-14-01944-f001:**
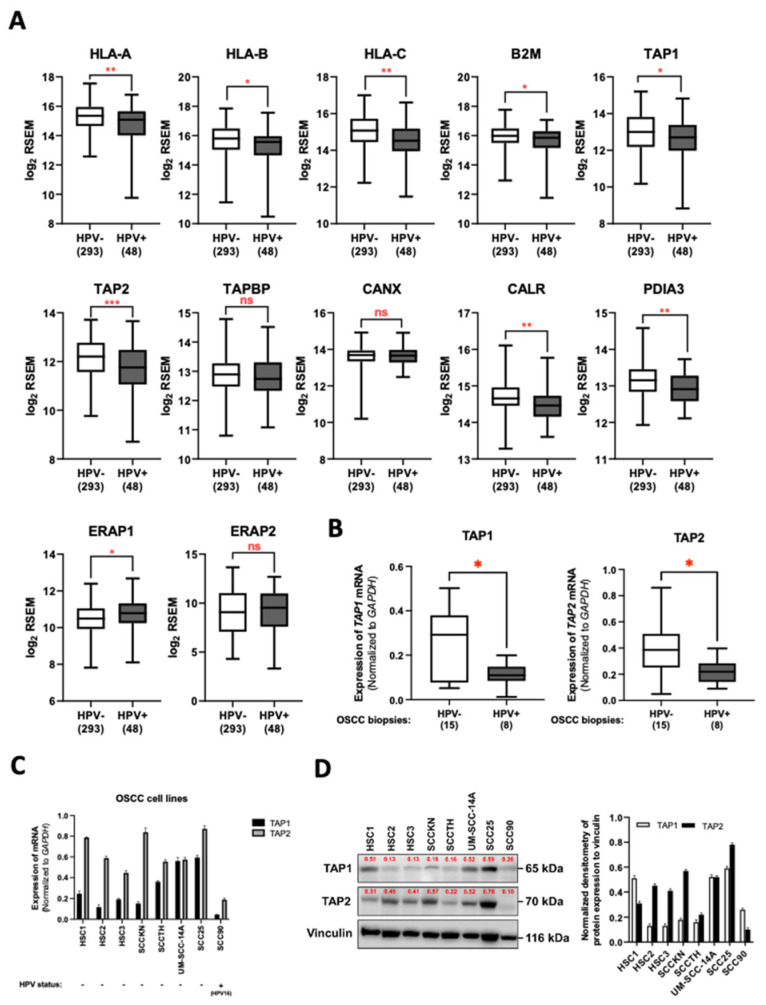
Expression of genes involved in MHC-I-dependent antigen presentation in OSCC samples of the HNSCC cohort from the TCGA database. (**A**) The expression of genes involved in the MHC-I heavy chain (HLA-A, HLA-B, and HLA-C), and other genes involved with MHC-I-dependent antigen presentation (B2M, TAP1, TAP2, TAPBP, CANX, CALR, PDIA3, ERAP1, and ERAP2) for HPV-positive and HPV-negative OSCC tissues. (**B**) The expression of TAP1 and TAP2 mRNA in OSCC biopsy samples with or without HPV infection was assessed by qRT-PCR. (C and D) The expression of TAP1 and TAP2 mRNA and proteins in the established OSCC cell lines was assessed by qRT-PCR (**C**) and immunoblotting (**D**). All experiments were carried out in triplicate, and at least three independent experiments were performed. The numbers in brackets refer to the number of samples included in each analysis. * *p* ≤ 0.05; ** *p* ≤ 0.01; *** *p* ≤ 0.001; ns—not significant.

**Figure 2 cancers-14-01944-f002:**
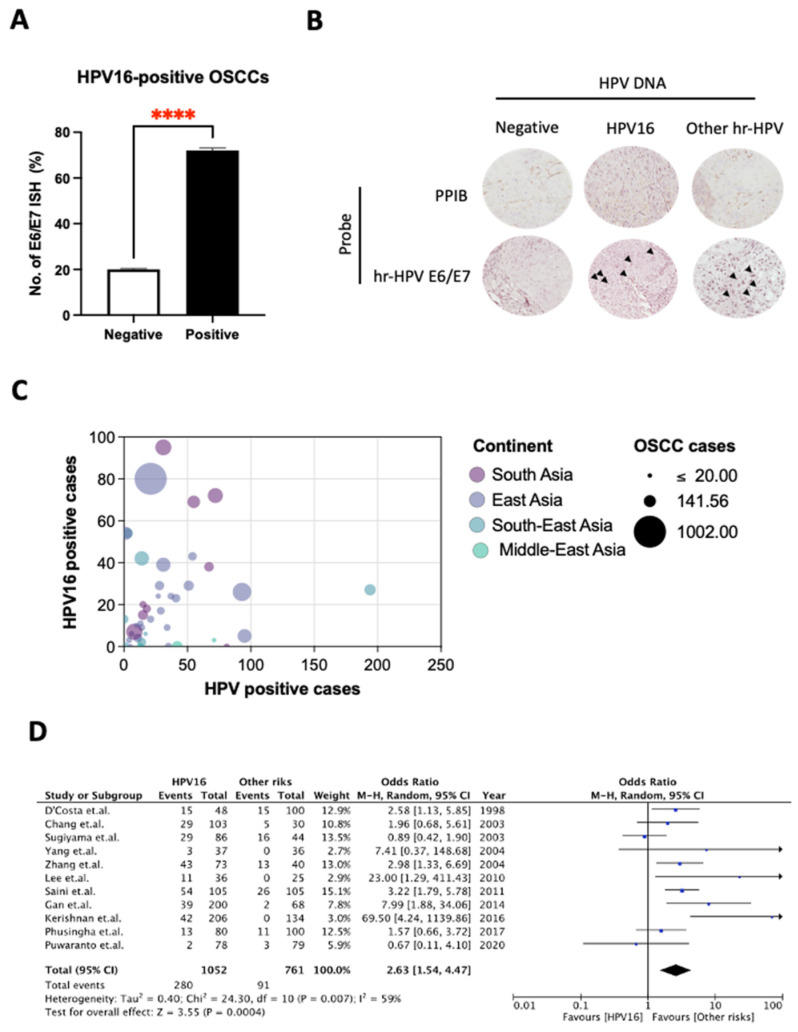
Comparison of presence of high-risk HPV among OSCC patients versus healthy individuals from different Asian studies. (**A**,**B**) The percentage of HPV16 in OSCC (FFPE) samples (*n* = 146), was assessed by E6/E7 mRNA ISH assay (RNAscope^®^ HPV kit, Advanced Cell Diagnostics Inc., Newark, CA, USA). All experiments were carried out in triplicate, and at least three independent experiments were performed. (**C**) The prevalence of HPV DNA and HPV16 in OSCC by geographical region in Asia. Filled circles correspond to study-specific prevalence. Sizes of filled circles and unfilled circles are proportional to the number of cases. (**D**) Forest plot of the association between HPV16 infection and OSCC risk. **** *p* ≤ 0.0001.

**Figure 3 cancers-14-01944-f003:**
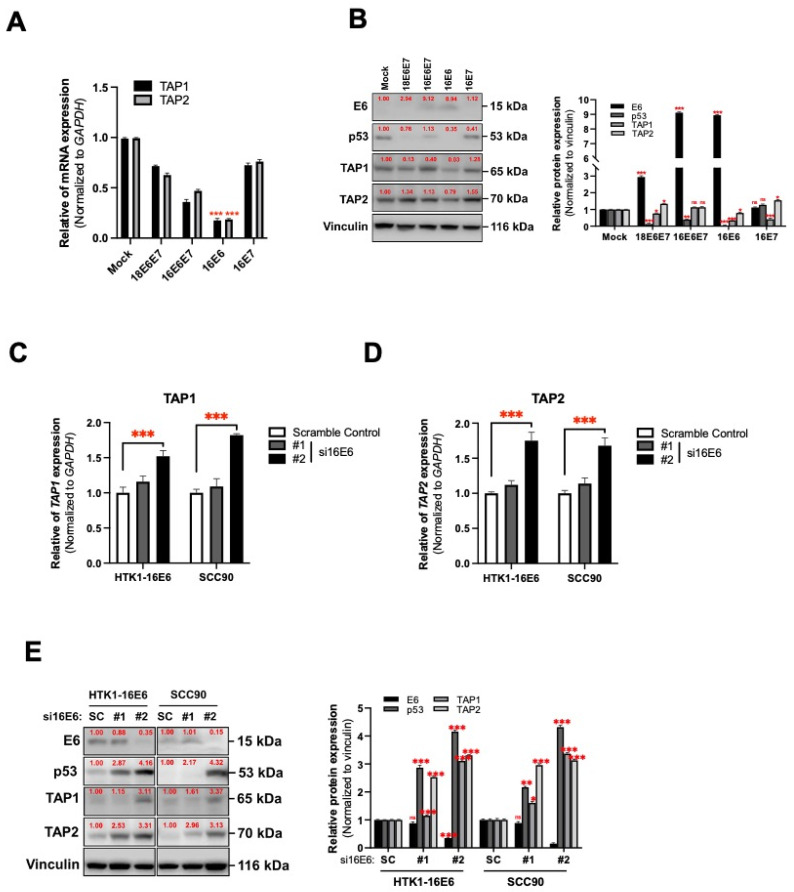
The E6 oncoprotein of HPV16 downregulates TAP1 and TAP2 expression in oral epithelial cells. (**A**,**B**) The expression of TAP1 and TAP2 mRNA and protein in HTK1-K4DT transduced with E6 and/or E7 from HPV16 or HPV18, assessed by qRT-PCR (**A**) and immunoblotting (**B**). (**C**–**E**) The expression of TAP1 and TAP2 mRNA and protein in HPV16 E6-expressing HTK1-K4DT transfected with siRNA-specific HPV16 E6, assessed by qRT-PCR ((**C**) for TAP1 and (**D**) for TAP2) and immunoblotting (**E**). All experiments were carried out in triplicate, and at least three independent experiments were performed. SC, scramble control. Red numbers indicate the relative intensity of the protein band as determined using ImageJ software. * *p* ≤ 0.05; ** *p* ≤ 0.01; *** *p* ≤ 0.001; ns—not significant.

**Figure 4 cancers-14-01944-f004:**
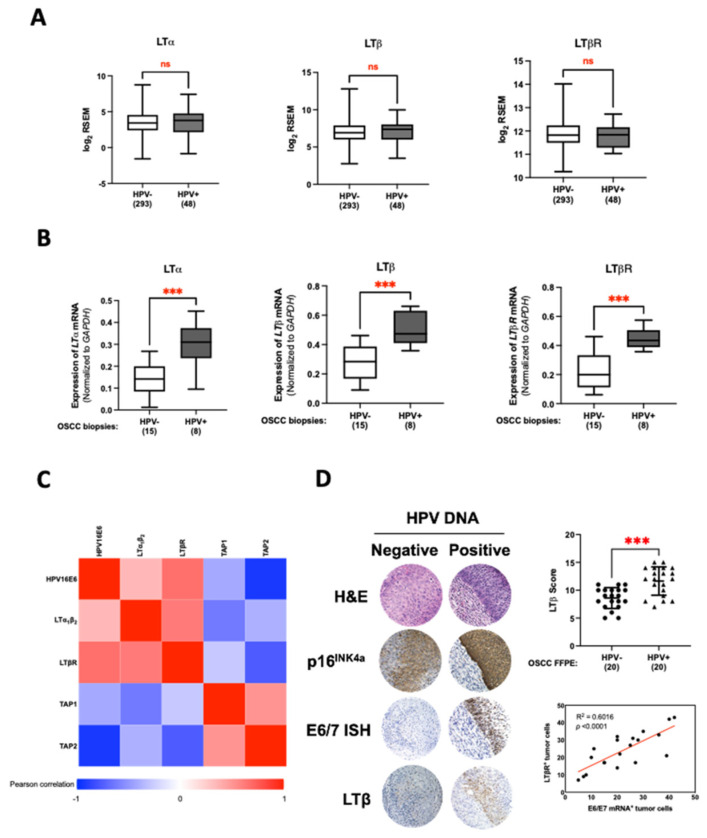
Involvement of the LT network in suppression of TAP1 and TAP2 in HPV-infected OSCCs. (**A**) The expression of LT-related genes (LTα, LTβ and LTβR) in HPV-positive and HPV-negative OSCC tissues. (**B**) The expression of LTα, LTβ, and LTβR mRNA in OSCC biopsy samples with HPV infection and non-HPV infection assessed by qRT-PCR. (**C**) Pearson correlation of mRNA expressions among LT-related genes (LTα, LTβ, and LTβR), HPV16E6, and TAP genes (TAP1 and TAP2) in OSCC tissues. Genes are shown on the horizontal and vertical axes. Red cells represent a positive correlation, and blue cells a negative correlation. The color bar indicates the r-value. (**D**) Serial sections of HPV DNA-negative and HPV DNA-positive OSCC tissues were stained with H&E, E6/E7 mRNA ISH, and antibodies targeting p16^INK4a^, and LTβ. All images were originally taken using a 20× objective lens, magnification: 400×. IHC scores of the indicated genes in 20 HPV-positive and 20 HPV-negative OSCC FFPE samples (right-upper). Pearson correlation was calculated for LTβR and HPV16 E6 (right-lower). All experiments were carried out in triplicate, and at least three independent experiments were performed. *** *p* ≤ 0.001; ns—not significant.

**Figure 5 cancers-14-01944-f005:**
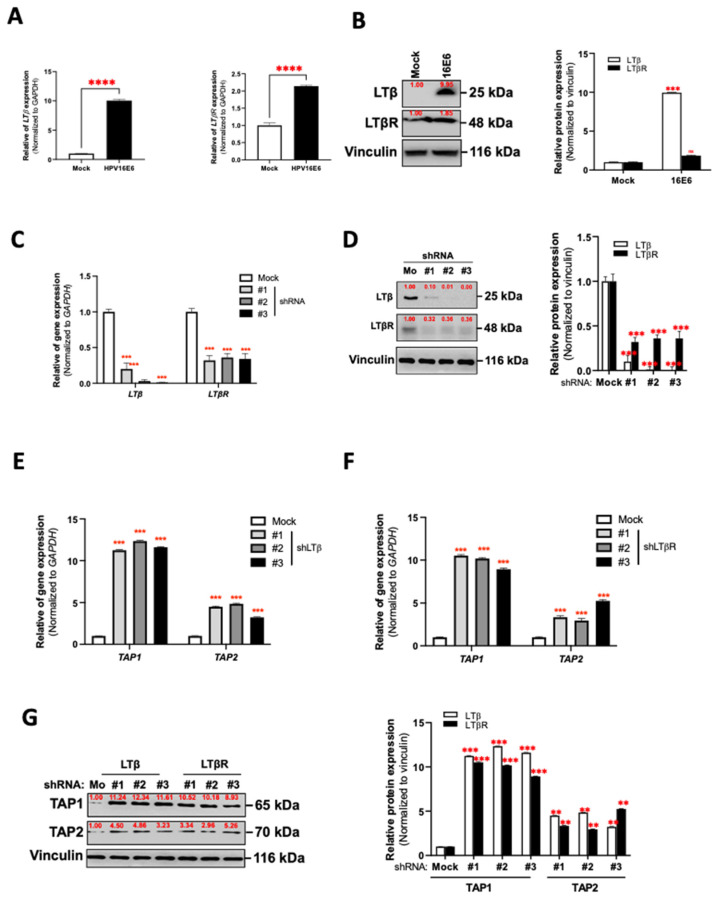
Role of LTβ and LTβR in suppression of TAP1 and TAP2 in HPV16 E6-expressing oral epithelial cells. (**A**,**B**) The expression of LTβ and LTβR mRNA and protein in HPV16 E6-expressing HTK1-K4DT was assessed by qRT-PCR (**A**) and immunoblotting (**B**). (**C**,**D**) The expression of LTβ and LTβR mRNA and protein in HPV16 E6-expressing HTK1-K4DT transfected with shRNA-specific LTβ, or LTβR, assessed by qRT-PCR (**C**) and immunoblotting (**D**). (**E**–**G**) The expression of TAP1 and TAP2 mRNA and protein in HPV16 E6-expressing HTK1-K4DT transfected with shRNA-specific LTβ, or LTβR, assessed by qRT-PCR ((**E**) for TAP1 and (**F**) for TAP2) and immunoblotting (**G**). All experiments were carried out in triplicate, and at least three independent experiments were performed. Red numbers indicate the relative intensity of the protein band according to ImageJ software. ** *p* ≤ 0.01; *** *p* ≤ 0.001; **** *p* ≤ 0.0001; #1, #2, and #3 indicated the number of LTβ, or LTβR-specific shRNA. ns—not significant.

**Figure 6 cancers-14-01944-f006:**
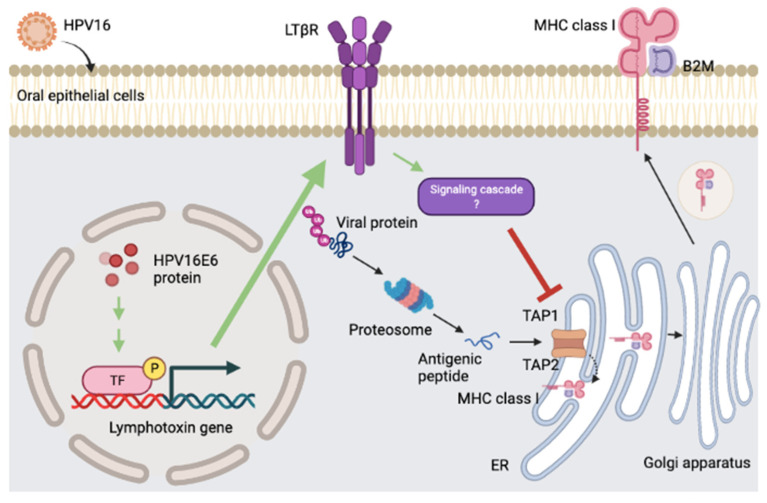
Proposed mechanisms of HPV16 E6 in oral carcinogenesis. HPV16 E6-activated LT is involved in TAP suppression in human oral cancer. HPV16 E6 upregulates lymphotoxin which functionally affects the stability of MHC-I by suppression of genes that are involved in MHC-I-dependent antigen presentation, TAP. Impaired T-cell responses could promote the persistent infection of HPV in oral cells, leading to oral cancer development and progression.

## Data Availability

The raw data supporting the conclusions of this manuscript will be made available by the authors, without undue reservation, to any qualified researcher.

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
