# Peer review of "Human Papillomavirus 16 E6 Suppresses Transporter Associated with Antigen-Processing Complex in Human Tongue Keratinocyte Cells by Activating Lymphotoxin Pathway"

_cancers, 2022, doi:10.3390/cancers14081944_

Round 1
Reviewer 1 Report
- “up-regulation of LTα, LTβ, and LTβR, was found in the HPV-positive OSCCs when compared to the HPV-negative group (Figure 4A).” However, the data shows no significance in Figure 4A.
- Why do the culture mediums contain ROCK inhibitor, TGF-βreceptor inhibitor, and 5% (v/v) Wnt3a and R-Spondin-1 for LWnt3a/RSPO1 cells?
- Expression of p16INK4Ahas highly correlated with HPV infection in head and neck squamous cell carcinoma. Why the intensity of p16 staining is strong in the HPV negative group?
- qRT-PCR and western data show the correlation between LT network and HPV16 E6. Is there direct evidence (protein - protein interaction) to confirm LT - E6 mechanism involved in HPV16 infection in OSCC?
Author Response
- “up-regulation of LTα, LTβ, and LTβR, was found in the HPV-positive OSCCs when compared to the HPV-negative group (Figure 4A).” However, the data shows no significance in Figure 4A.
Response: Thank you for kindly providing the imperative point for this figure. We retrieved level 3 mRNA expression of LTα, LTβ, and LTβR genes and analyzed them against their HPV status. As shown in Figure 4A, the mRNA expression of such genes showed no statistically significant difference in HPV positive cases when compared to HPV negative cases. However, we found an increasing trend of these genes in the HPV-positive group relative to others. This result has been rewritten to “an increasing trend of LTα, LTβ, and LTβR could be found in the HPV-positive OSCCs when compared to the HPV-negative group (Figure 4A)”. Please see this correction in the revised main text (Line 442-443).
- Why do the culture mediums contain ROCK inhibitor, TGF-βreceptor inhibitor, and 5% (v/v) Wnt3a and R-Spondin-1 for LWnt3a/RSPO1 cells?
Response: HTK1 cell line was used as a model in this study. It was derived from normal tongue tissue without pathological lesions. F-medium supplemented with Y-27632 (ROCK inhibitor) and 5% FBS was used to culture this cell line. Thus, to avoid the effect of FBS on keratinocytes differentiation, enhance proliferation, and keep cell characteristics, we further supplemented this cultured medium with a TGF-β receptor inhibitor, and 5% (v/v) Wnt3a and R-Spondin-1. To make it for a better understanding of cell cultivation, we delete “for LWnt3a/RSPO1 cells”. Please see the revised version (Line 141).
- Expression of p16INK4Ahas highly correlated with HPV infection in head and neck squamous cell carcinoma. Why the intensity of p16 staining is strong in the HPV negative group?
Response: Thank you for kindly providing the crucial point. We raise your perception here. Although the overexpression of p16INK4A has been proposed to use as one of the surrogate markers for HPV infection particularly, in cervical cancer and oropharyngeal carcinomas, there had been substantial differences occur in the means of HPV infection and indicative overexpression of p16INK4A amongst oral carcinoma (OSCC) studies. The p16INK4A gene is frequently mutated in oral cancer (60%) that was affected by various factors such as both the patient’s genetic background and behavioral environments. In this condition, p16INK4A overexpression might be positive without HPV infection (Zhao et. al., 2016). In the context of HPV infection, OSCC has a lower positive rate of E6 and E7 mRNA expression compared with that HPV DNA positive rate. It has been hypothesized that there were two stages of HPV DNA-positive OSCCs. One is the latent infection that limited the expression of E6 and E7, caused by the HPV DNA integration form. The other is an active infection in non-keratinized OSCC that was the overexpression of such oncogenes, triggered by the HPV nuclear episome and/or genome integration. (Ndiaye et. al., 2014; Tokuzen et. al., 2021). However, the systemic review of Grønhøj Larsen et. al., 2014 on the correlation between HPV and p16INK4A overexpression in oropharyngeal tumors also suggested the cutoff of p16INK4A as ⩾70% of cytoplasmic and nuclear staining. Therefore, the p16INK4A signal might be found in HPV-negative OSCC tissue.
- qRT-PCR and western data show the correlation between LT network and HPV16 E6. Is there direct evidence (protein-protein interaction) to confirm the LT - E6 mechanism involved in HPV16 infection in OSCC?
Response: Thank you for kindly providing the interesting issue for our work. In this study, we aimed to demonstrate the effect of HPV16 E6 on the MHC-I-dependent genes, TAP1 and TAP2. Notably, our results indicated the upregulation of the lymphotoxin genes was a central network, involved in HPV16 E6-mediated TAP1 and TAP2 suppression. Although the correlation of HPV16 E6 and lymphotoxin genes has been studied by several reports (Kim et. al., 2008; Liu et. al., 2020), there are limited pieces of evidence of the protein-protein interaction to confirm such mechanisms involved in HPV16-infected OSCC. Therefore, it is a great chance for us to consider and design a further study.
References:
Zhao, R., Choi, B. Y., Lee, M. H., Bode, A. M., & Dong, Z. (2016). Implications of Genetic and Epigenetic Alterations of CDKN2A (p16(INK4a)) in Cancer. EBioMedicine, 8, 30–39. https://doi.org/10.1016/j.ebiom.2016.04.017
Ndiaye, C., Mena, M., Alemany, L., Arbyn, M., Castellsagué, X., Laporte, L., Bosch, F. X., de Sanjosé, S., & Trottier, H. (2014). HPV DNA, E6/E7 mRNA, and p16INK4a detection in head and neck cancers: a systematic review and meta-analysis. The Lancet. Oncology, 15(12), 1319–1331. https://doi.org/10.1016/S1470-2045(14)70471-1
Tokuzen, N., Nakashiro, K. I., Tojo, S., Goda, H., Kuribayashi, N., & Uchida, D. (2021). Human papillomavirus-16 infection and p16 expression in oral squamous cell carcinoma. Oncology letters, 22(1), 528. https://doi.org/10.3892/ol.2021.12789
Grønhøj Larsen, C., Gyldenløve, M., Jensen, D. H., Therkildsen, M. H., Kiss, K., Norrild, B., Konge, L., & von Buchwald, C. (2014). Correlation between human papillomavirus and p16 overexpression in oropharyngeal tumours: a systematic review. British journal of cancer, 110(6), 1587–1594. https://doi.org/10.1038/bjc.2014.42
Kim, D. H., Kim, E. M., Lee, E. H., Ji, K. Y., Yi, J., Park, M., Kim, K. D., Cho, Y. Y., & Kang, H. S. (2011). Human papillomavirus 16E6 suppresses major histocompatibility complex class I by upregulating lymphotoxin expression in human cervical cancer cells. Biochemical and biophysical research communications, 409(4), 792–798. https://doi.org/10.1016/j.bbrc.2011.05.090
Liu, Y. H., Chen, H. L., Xu, B. Q., Wei, K., & Ying, X. Y. (2020). A preliminary study on the immune responses of HPV16-E7 by combined intranasal immunization with lymphotoxin. Ginekologia polska, 91(6), 301–307. https://doi.org/10.5603/GP.2020.0055
Reviewer 2 Report
This paper highlighted how impaired T-cell responses could promote the persistent infection of HPV in oral cells, leading to oral cancer development and progression demonstrating that HPV16 E6-activated lymphotoxin pathway is involved in TAP suppression in human oral cancer.Minor point: - Please replace HNSC with HNSCC in all the text. Major points: - The E6 and E7 specific knockdown experiments are somewhat surprising given that E6 and E7 are expressed from a polycistronic RNA. It would be appropiate that the authors will provide details about the siRNAs used. It is very important to demonstrated that this siRNA doesn't affect E7 expression.
Author Response
This paper highlighted how impaired T-cell responses could promote the persistent infection of HPV in oral cells, leading to oral cancer development and progression demonstrating that the HPV16 E6-activated lymphotoxin pathway is involved in TAP suppression in human oral cancer.
Minor point: - Please replace HNSC with HNSCC in all the text.
Response: Thank you for kindly providing this suggestion for our work, we wrote HNSCC for the clarification of “Head and Neck squamous cell carcinoma”. However, it is better to replace the abbreviation with HNSC in this context. We agree on the “HNSC” term as you mentioned above and fix it all in the revised version. Please consider.
Major points: - The E6 and E7 specific knockdown experiments are somewhat surprising given that E6 and E7 are expressed from a polycistronic RNA. It would be appropriate that the authors will provide details about the siRNAs used. It is very important to demonstrate that this siRNA doesn't affect E7 expression.
Response: Thank you for providing the critical information. We appreciate your notion here. To determine the effect of E6 on the TAP1 and TAP2 expression, the assay of E6-specific siRNA was performed. In the main text, the results showed that knocked-down E6 restored the expression of TAP1 and TAP2 in both mRNA and protein as we declared in figure 3C-3E. However, the potential and specificity of the designed E6-specific siRNA were firstly assessed. According to the previous publications (Yamato et. al, 2008; Chang et. al, 2010), we designed several siRNAs and tested them. Using qRT-PCR, the best 2 siRNAs were selected and used for the downstream experiments. We have mentioned their sequences in Supplement Table 5. The selected siRNAs that affected HPV16 E6 expression were shown in Supplementary Figure 2B. Furthermore, we consequently evaluated whether HPV16 E7 was not significantly altered by the selected siRNAs. Thus, the overall results of the selected siRNAs on HPV16 E6 in potential and specificity-dependent manner were fixed and shown in Supplementary Figure 2B in the revised version. Please consider.
References:
Yamato, K., Yamada, T., Kizaki, M., Ui-Tei, K., Natori, Y., Fujino, M., Nishihara, T., Ikeda, Y., Nasu, Y., Saigo, K., & Yoshinouchi, M. (2008). New highly potent and specific E6 and E7 siRNAs for treatment of HPV16 positive cervical cancer. Cancer gene therapy, 15(3), 140–153. https://doi.org/10.1038/sj.cgt.7701118
Chang, J. T., Kuo, T. F., Chen, Y. J., Chiu, C. C., Lu, Y. C., Li, H. F., Shen, C. R., & Cheng, A. J. (2010). Highly potent and specific siRNAs against E6 or E7 genes of HPV16- or HPV18-infected cervical cancers. Cancer gene therapy, 17(12), 827–836. https://doi.org/10.1038/cgt.2010.38
Reviewer 3 Report
Manuscript ID: Cancers-1648553
Title: “Human Papillomavirus 16 E6 Suppresses Transporter Associated with Antigen-Processing Complex in Human Tongue Keratinocyte Cells by Activating Lymphotoxin Pathway’’
In the manuscript, Burassakarn et al. aimed to shed light on HPV16 oncogenic role in oral squamous cell carcinomas (OSCCs), the main subtype of HNSCs. The authors show that HPV16 E6 promotes LTα1β2 and LTβR expression, thus promoting the lymphotoxin signaling pathway that leads to suppression of the transporter associated with the antigen-processing complex (TAPs; TAP1 and TAP2). Through analysis of the transcriptomic profiles in the head-and-neck squamous cell carcinoma (HNSC) cohort from the TCGA database and further validation in fresh clinical samples, the authors also demonstrated regulation of the antigenic peptide-loaded machinery in HPV-infected OSCC tissues. Finally, the authors conclude that HPV16 E6-associated immune-evasive properties in oral epithelial cells play a functional role in the development of HPV-mediated OSCCs.
The manuscript is well written. The subject of the study is important as detailed mechanisms directly involved in the suppression of MHC I expression in OSCC are not clearly understood. Moreover, the presented data clinically corroborate with previous in vitro studies in elucidating the underlying mechanism of immune evasion and HPV E6 contribution to the clinical course of OSCC development. However, I have a few concerns that should be addressed to create a stronger paper that more clearly demonstrates its “innovation” in the field prior to publication.
Comments:
- It is intriguing to see that overall MHC-I-dependent antigen presentation genes are high in HNSC than normal but low in the HPV+ve TCGA cohort. However, the in vitro cell line data for TAP1 and TAP2 in Figure 1C-D show some inconsistencies with TCGA expression and HPV correlation. Are these cells derived from pre-malignant tumor biopsies?
- In relation to the above comment, does the variable expression of TAP1 and TAP2 in these cell lines infer due to HPV16 E6 physical state or low copy number in the absence of other confounding variables?
- Do HPV16-E6+ve cells with low TAP1/2 expression modulate cytotoxic T lymphocytes (CTLs) mediated tumor surveillance? Please provide data with proper controls.
- What is the status of p53 and Rb in SCC90 oral cancer cell line and how does it correlate with TAP1/2 expression?
- In Figure 4A, the author stated up-regulation of LTα, LTβ, and LTβR, in HPV-positive OSCCs compared to HPV-negative and a similar statement is provided for Figure 4B. However, Figure 4A data show no significant up-regulation despite having a good number of OSCC cases while Figure 4B show significant up-regulation with a low number of cases. Please discuss the discrepancy and reframe the statement as per the data.
- In the context of the immune-evasive mechanism, how do HPV+ve oral cancer cells respond to IFN-γ and TNF-α stimulation, and what is the level of MHC-I dependent, LTβR expression in this condition?
- Information about how many replicates per experiment were analyzed is missing, making it difficult to assess statistical validity.
- Majority of the experiment reported in the manuscript was carried out using only one cell line, HTK1 to set up the scientific rationale. Although validation is shown in a bunch of cell lines. However, to increase scientific rigor and reproducibility, it is advisable to repeat the key experiment with another cell line to conclude clinical findings.
- The resolution of IHC image is of poor quality and not clear enough to interpret the findings. Please provide better resolution, contrast, and large-size images with proper quantification.
- To correlate lymphotoxin signaling in mediating HPV16E6 downregulated MHC I dependent gene, it's imperative to include recombinant lymphotoxin in vitro data for key regulatory factors and CTL activity to correlate the findings clinically.
Author Response
- It is intriguing to see that overall MHC-I-dependent antigen presentation genes are high in HNSC than normal but low in the HPV+ve TCGA cohort. However, the in vitro cell line data for TAP1 and TAP2 in Figure 1C-D show some inconsistencies with TCGA expression and HPV correlation. Are these cells derived from pre-malignant tumor biopsies?
Response: As described by the original publications, these cells were established and derived from cancer lesions, not derived from pre-malignant tumors. The origin of these cells was following;
No. |
Cell name |
Origin |
1 |
HSC1 |
Oral cavity |
2 |
HSC2 |
Oral cavity: floor of mouth |
3 |
HSC3 |
Tongue |
4 |
SCCKN |
Tongue |
5 |
SCC90 (HPV16+ cells) |
Tongue |
6 |
SCC25 |
Tongue |
7 |
SCCTH |
Oral cavity |
8 |
UM-SCC-14A |
Oral cavity: floor of mouth |
- In relation to the above comment, does the variable expression of TAP1 and TAP2 in these cell lines infer due to HPV16 E6 physical state or low copy number in the absence of other confounding variables?
Response: For these oral cancer cell lines, only SCC90 contained a naturally integrated HPV16 genome from 100-150 copies. Compared to SiHa and CaSki, this cell contained more E6 copy numbers (approximately 40 copies) than SiHa (approximately 10 copies) and CaSki (approximately 20 copies) (Ferris et. al., 2005). Thus, the variable expression of TAP1 and TAP2 in SCC90 might not be directly affected by the physical state or low copy number of HPV16 E6. We assumed the differences in genetic landscape amongst the cells and the disruption of cellular genes by HPV integrants are the possible confounding factors to vary TAP1 and TAP2 expression in these cells. However, further experiment is necessitated to clarify this point.
- Do HPV16-E6+ve cells with low TAP1/2 expression modulate cytotoxic T lymphocytes (CTLs) mediated tumor surveillance? Please provide data with proper controls.
Response: Yes, the expression of HPV16E6 might be strongly enhanced HTK1 resistance to cytotoxic T lymphocytes (CTLs)-mediated lytic activity, and knockdown of HPV16 E6 by antisense could have the opposite effect. As described by Kim et. al., 2011, CTL activity was determined by an LDH releasing assay at the indicated effector: target (E: T) ratios in HeLa and Caski cells stably expressing HPV16E6 and antisense HPV16E6, respectively and the results suggested that HPV16E6 plays a key role in the ability of CTL-mediated tumor surveillance via the downregulation of MHC-I expression. Since the expression of TAP1 and TAP2 were directly linked to the antigenic presentation and MHC-I function, therefore, we assumed that the expression of HPV16E6 might be associated with an impaired capacity of the cells to respond to cytotoxic T lymphocytes (CTLs)-mediated lytic activity.
- What is the status of p53 and Rb in the SCC90 oral cancer cell line and how does it correlate with TAP1/2 expression?
Response: As described by Kalu et. al., 2017, a wild-type of p53 and Rb were observed in the SCC90. Compared to other oral cancer cell lines, SCC90 expressed lower p53 as well as TAP1 and TAP2. It could be proposed that p53 regulates endogenous antigen presentation through transcriptional control of ERAP1 and TAP1 (Blagih et. al., 2020). TAP1 expression is enhanced through p53-mediated transcription in response to DNA damage or direct activation of p53, leading to increased surface MHC-I–peptide complexes in cancer cells. p53-dependent induction of ERAP1 expression (Wang et al., 2013) would also enhance the number of peptides available for MHC-I loading. Both these aspects of antigen presentation are downregulated in p53-mutant and p53-null cell lines (Wang et al., 2013; Zhu et al., 1999). Intriguingly, deletion of key components of the MHC-I pathway (i.e., β2M or TAP1) reduces p53 function, suggesting an interplay between the MHC-I presentation pathway and p53 activity in cancer cells that are not yet fully understood (Sabapathy et. al., 2008).
- In Figure 4A, the author stated up-regulation of LTα, LTβ, and LTβR, in HPV-positive OSCCs compared to HPV-negative and a similar statement is provided for Figure 4B. However, Figure 4A data show no significant up-regulation despite having a good number of OSCC cases while Figure 4B show significant up-regulation with a low number of cases. Please discuss the discrepancy and reframe the statement as per the data.
Response: Thank you for kindly providing the imperative point for these figures. We retrieved level 3 mRNA expression of LTα, LTβ, and LTβR genes and analyzed them against their HPV status. As shown in Figure 4A, the mRNA expression of such genes showed no statistically significant difference in HPV positive cases when compared to HPV negative cases. However, we found an increasing trend of these genes in the HPV positive group relative to others. It is better to rewrite that “an increasing trend of LTα, LTβ, and LTβR, could be found in the HPV-positive OSCCs when compared to the HPV-negative group (Figure 4A)” instead of the previous one. For the TCGA database, they conducted their analysis with the large numbers of tissue samples of head and neck cancer patients from the various regional cancer centers, globally. Therefore, the differences in geography and ethnicity of patients are one explanation for the varied HPV-mediated LTα, LTβ, and LTβR up-regulation. However, inconsistencies in the HPV prevalence in the oral cavity, in particular, HPV-positive oropharyngeal carcinomas, have obscured the ability to definitively associate HPV with OSCC. In addition, non-keratinized OSCC exhibiting high E6 and/or E7 mRNA expression shows a more increased LTα, LTβ, and LTβR expression than the others, suggesting that LTα, LTβ, and LTβR mRNA expression may be varied in assigning HPV-positive OSCC (Kim et. al., 2008; Liu et. al., 2020). In Figure 4B, the expression of such genes was evaluated from our previous tissue OSCC samples that were retrieved from Southeast Asia’s patients and showed high E6/E7 mRNA levels (Phusingha et. al., 2017). Accordingly, it might be observed that HPV16 infection is statistically associated with an increased expression of MHC-I-dependent genes in OSCCs. However, additional high-quality studies with larger sample sizes are needed to further confirm such relationships. The overall discussions have been stated more in the discussion part of the revised version. Please see (Line 564-578).
- In the context of the immune-evasive mechanism, how do HPV+ve oral cancer cells respond to IFN-γ and TNF-α stimulation, and what is the level of MHC-I dependent, LTβR expression in this condition?
Response: Although OSCC has a lower positive rate of E6 and E7 mRNA expression compared with that HPV DNA positive rate, the presence of high-risk HPV might be associated with an impaired capacity of cells to respond to IFN-γ and TNF-α, asshown by the lower messenger RNA (mRNA) expression and production of the IFN-γ and/or TNF-α-induced pro-inflammatory cytokines CCL2, RANTES (CCL5), interleukin (IL)-8, and the chemokines CXCL9, 10 and 11 by oral cancer cells. The high-risk HPV-mediated deregulated expression of STAT1 (Nees et. al., 2001; Zhou et. al., 2013) may explain the impaired cytokine expression by high-risk HPV-positive oral cancer cells upon IFN-γ stimulation, but not the impaired response to TNF-α (IL-8) or IFN-γ and TNF-α (RANTES). Moreover, high-risk HPV hampers phosphorylation of the NF-kB subunit RelA (p65) upon stimulation with the rapid TNF-α induction (Tummers et. al, 2015). Since high-risk HPV impaired the cell’s capacity to respond to such cytokines, thus the expression level of LTβR was down-regulated in this condition.
- Information about how many replicates per experiment were analyzed is missing, making it difficult to assess statistical validity.
Response: Thank you for kindly providing the important point for our work. This study performed at least three replicates with the same batch and three independent experiments to verify the results between batches. The statement “All experiments were carried out in triplicate, and at least three independent experiments were performed” has been indicated in the revised version. Please see (Line 305-306).
- The majority of the experiment reported in the manuscript was carried out using only one cell line, HTK1 to set up the scientific rationale. Although validation is shown in a bunch of cell lines. However, to increase scientific rigor and reproducibility, it is advisable to repeat the key experiment with another cell line to conclude clinical findings.
Response: Thank you for kindly providing the critical point for this work. We appreciated your conception here. An infection of high-risk HPVs particularly, HPV16 is one of the major factors for carcinogenesis of the anogenital tract as well as oropharyngeal carcinomas. Although carcinomas in the oral cavity account for the 2nd rank (up to 30%) of HPV-associated head and neck cancers, progress in examining HPV’s role in their pathogenesis has been limited by the difficulty in establishing latent infection in non-transformed oral epithelial cells. Since HPV infection of primary epithelial cells typically results in productive infection (Leemans et. al, 2018), the study of latent HPV infection of cultured oral carcinoma cell lines has been largely limited. Additionally, these cell lines are of limited utility in understanding the role of latent HPV in the development of carcinomas via immune suppression because they cannot model critical premalignant virus-host interactions. According to our previous study (Phusingha et. al., 2017), the highest prevalence of HPV infection in OSCC was observed in the tongue and statistically associated with HPV16. This result was consistently reported by other studies. We, therefore, conducted this work only with primarily non-transformed tongue cell lines, the well-established non-transformed tongue cell lines from human tongue keratinocytes (HTK) as we mentioned in the main text. Although, recent human telomerase reverse transcriptase (hTERT)-immortalized normal oral keratinocytes (NOKs) have emerged as a model that summarizes aspects of EBV and HPV infection in vitro and in vivo (Guidry et. al., 2018), however, these cells were derived from gingival tissues (Piboonniyom et. al., 2003).
- The resolution of the IHC image is of poor quality and not clear enough to interpret the findings. Please provide better resolution, contrast, and large-size images with proper quantification.
Response: Thank you for kindly providing the focal point for this figure. We gain your notion here. According to the guideline from the Cancers, the quality of such figures has been improved.
- To correlate lymphotoxin signaling in mediating HPV16E6 downregulated MHC I dependent gene, it's imperative to include recombinant lymphotoxin in vitro data
Response: Thank you for kindly providing the essential point for this work. We appreciated your conception here. In recent work, we newly described the role of HPV16 E6 in the underlying mechanism of oral carcinogenesis via MHC-I-dependent gene suppression. According to the qualified literacy that showed the overexpression of lymphotoxin genes affected the expression of MHC-I genes in HPV16 and/or HPV18-infected cervical cancer cells (Kim et al., 2011), we, therefore, focused on such partway and however decided to perform the chase lymphotoxin genes experiment using shRNA only as mentioned in the main text.
References:
Phusingha, P., Ekalaksananan, T., Vatanasapt, P., Loyha, K., Promthet, S., Kongyingyoes, B., Patarapadungkit, N., Chuerduangphui, J., & Pientong, C. (2017). Human papillomavirus (HPV) infection in a case-control study of oral squamous cell carcinoma and its increasing trend in northeastern Thailand. Journal of medical virology, 89(6), 1096–1101. https://doi.org/10.1002/jmv.24744
Guidry, J. T., Myers, J. E., Bienkowska-Haba, M., Songock, W. K., Ma, X., Shi, M., Nathan, C. O., Bodily, J. M., Sapp, M. J., & Scott, R. S. (2019). Inhibition of Epstein-Barr Virus Replication in Human Papillomavirus-Immortalized Keratinocytes. Journal of virology, 93(2), e01216-18. https://doi.org/10.1128/JVI.01216-18
Piboonniyom, S. O., Duensing, S., Swilling, N. W., Hasskarl, J., Hinds, P. W., & Münger, K. (2003). Abrogation of the retinoblastoma tumor suppressor checkpoint during keratinocyte immortalization is not sufficient for induction of centrosome-mediated genomic instability. Cancer research, 63(2), 476–483.
Leemans, C. R., Snijders, P., & Brakenhoff, R. H. (2018). The molecular landscape of head and neck cancer. Nature reviews. Cancer, 18(5), 269–282. https://doi.org/10.1038/nrc.2018.11
Nees, M., Geoghegan, J. M., Hyman, T., Frank, S., Miller, L., & Woodworth, C. D. (2001). Papillomavirus type 16 oncogenes downregulate the expression of interferon-responsive genes and upregulate proliferation-associated and NF-kappaB-responsive genes in cervical keratinocytes. Journal of virology, 75(9), 4283–4296. https://doi.org/10.1128/JVI.75.9.4283-4296.2001
Zhou, F., Chen, J., & Zhao, K. N. (2013). Human papillomavirus 16-encoded E7 protein inhibits IFN-γ-mediated MHC class I antigen presentation and CTL-induced lysis by blocking IRF-1 expression in mouse keratinocytes. The Journal of general virology, 94(Pt 11), 2504–2514. https://doi.org/10.1099/vir.0.054486-0
Tummers, B., Goedemans, R., Pelascini, L. P., Jordanova, E. S., van Esch, E. M., Meyers, C., Melief, C. J., Boer, J. M., & van der Burg, S. H. (2015). The interferon-related developmental regulator 1 is used by human papillomavirus to suppress NFκB activation. Nature communications, 6, 6537. https://doi.org/10.1038/ncomms7537
Ferris, R. L., Martinez, I., Sirianni, N., Wang, J., López-Albaitero, A., Gollin, S. M., Johnson, J. T., & Khan, S. (2005). Human papillomavirus-16 associated squamous cell carcinoma of the head and neck (SCCHN): a natural disease model provides insights into viral carcinogenesis. European journal of cancer (Oxford, England : 1990), 41(5), 807–815. https://doi.org/10.1016/j.ejca.2004.11.023
Kalu, N. N., Mazumdar, T., Peng, S., Shen, L., Sambandam, V., Rao, X., Xi, Y., Li, L., Qi, Y., Gleber-Netto, F. O., Patel, A., Wang, J., Frederick, M. J., Myers, J. N., Pickering, C. R., & Johnson, F. M. (2017). Genomic characterization of human papillomavirus-positive and -negative human squamous cell cancer cell lines. Oncotarget, 8(49), 86369–86383. https://doi.org/10.18632/oncotarget.21174
Blagih, J., Buck, M. D., & Vousden, K. H. (2020). p53, cancer and the immune response. Journal of cell science, 133(5), jcs237453. https://doi.org/10.1242/jcs.23745
Wang, B., Niu, D., Lai, L., & Ren, E. C. (2013). p53 increases MHC class I expression by upregulating the endoplasmic reticulum aminopeptidase ERAP1. Nature communications, 4, 2359. https://doi.org/10.1038/ncomms3359
Zhu, K., Wang, J., Zhu, J., Jiang, J., Shou, J., & Chen, X. (1999). p53 induces TAP1 and enhances the transport of MHC class I peptides. Oncogene, 18(54), 7740–7747. https://doi.org/10.1038/sj.onc.1203235
Sabapathy, K., & Nam, S. Y. (2008). Defective MHC class I antigen surface expression promotes cellular survival through elevated ER stress and modulation of p53 function. Cell death and differentiation, 15(9), 1364–1374. https://doi.org/10.1038/cdd.2008.55
Kim, D. H., Kim, E. M., Lee, E. H., Ji, K. Y., Yi, J., Park, M., Kim, K. D., Cho, Y. Y., & Kang, H. S. (2011). Human papillomavirus 16E6 suppresses major histocompatibility complex class I by upregulating lymphotoxin expression in human cervical cancer cells. Biochemical and biophysical research communications, 409(4), 792–798. https://doi.org/10.1016/j.bbrc.2011.05.090
Liu, Y. H., Chen, H. L., Xu, B. Q., Wei, K., & Ying, X. Y. (2020). A preliminary study on the immune responses of HPV16-E7 by combined intranasal immunization with lymphotoxin. Ginekologia polska, 91(6), 301–307. https://doi.org/10.5603/GP.2020.0055
Round 2
Reviewer 2 Report
The authors said that they "...wrote HNSCC for the clarification of “Head and Neck squamous cell carcinoma”" but this is not true. In the new version of the text they didn't replace HNSC with HNSCC. They wrote "oral squamous cell carcinomas (OSCCs)", just why SCC is the abbreviation for "squamous cell carcinoma".Statistical analysis is missing in several figures (even supplementary figures), please provide.
Author Response
The authors said that they "...wrote HNSCC for the clarification of “Head and Neck squamous cell carcinoma” but this is not true. In the new version of the text, they didn't replace HNSC with HNSCC. They wrote "oral squamous cell carcinomas (OSCCs)", just why SCC is the abbreviation for "squamous cell carcinoma".
Response: Thank you for kindly providing this suggestion again, we realize your impression here and apologize for our technical error. In the new version, all HNSC wording has been replaced with “HNSCC”. Please find them.
Statistical analysis is missing in several figures (even supplementary figures), please provide.
Response: Thank you for kindly providing the imperative point for these figures. We have added the p-value of statistical analysis in the term asterisk sign to all figures in the main text particularly, the densitometry graphs of protein expressions and even the Supplementary Figure 2. Please see them in the new version.

Reviewer 3 Report
My concerns and questions have been adequately addressed. Simply include the source of all cell lines used in the Material and Method section, along with proper citations.
Author Response
My concerns and questions have been adequately addressed. Simply include the source of all cell lines used in the Material and Method section, along with proper citations.
Response: Thank you for your all suggestions and critical points to our present work. We appreciate it. The source of all cell lines that we used in this study has been addressed in Supplementary Table 2. Please find it.
